# Analysis of Neural Interface When Using Modiolar Electrode Stimulation. Radiological Evaluation, Trans-Impedance Matrix Analysis and Effect on Listening Effort in Cochlear Implantation

**DOI:** 10.3390/jcm10173962

**Published:** 2021-08-31

**Authors:** Angel Ramos-de-Miguel, Juan Carlos Falcón-González, Angel Ramos-Macias

**Affiliations:** 1Hearing and Balance Laboratory, Las Palmas de Gran Canaria University (SIANI), 35001 Las Palmas, Spain; ramosorl@idecnet.com; 2Department of Otolaryngology, Head and Neck Surgery, Complejo Hospitalario Universitario Insular Materno Infantil de Gran Canaria, 35016 Las Palmas, Spain; jfalgond@gobiernodecanarias.org

**Keywords:** cochlear implant, trans-impedance matrix, pupillometry, electrode discrimination

## Abstract

Background: The proximity of the electrode to the modiolar wall may be of interest to investigate the effect of pitch discrimination. This research establishes the relation between these factors and whether perimodiolar positions may provide benefits regarding improved electrode discrimination. Methods: A prospective randomized study including 24 post-lingual deaf adults was performed. A psychoacoustic study was done by using a psychoacoustic research platform. Radiological study, and a cone-beam computed tomography was used to assess post cochlear implantation electrodes’ position. Trans-impedance matrix (TIM) analysis was performed after cochlear implant insertion in all cases, and pupillometry test was also performed. Results: 12 patients received a slim perimodiolar electrode array, and 12 patients received a straight electrode array. Although all the patients showed similar speech test results after 12 months follow-up, those implanted with a perimodiolar electrode obtained better scores in electrode discrimination test and pupillometry test, and showed more homogenous TIM patterns. Conclusions: The better positioning of the electrode array seams to provide a better hearing resolution and less listening effort trans-impedance matrix seems to be a useful tool to analyze positioning of the perimodiolar array.

## 1. Introduction

Different studies have shown significant correlations between electrode discrimination and speech perception in cochlear implant (CI) users [1,2,3,4,5]. The electrode discrimination test is based on the patient’s ability to distinguish the pitch generated by two different electrodes. Channel interaction is still a difficult topic, despite significant improvements in hardware and signal processing have been made due to technical advances [6,7,8].

Impedance values may be related to cochlear fibrosis, but this is not the only reason by which impedance differences can occur. Impedance differences between perimodiolar and straight electrodes after implantation may also be related to electrode distance from the modiolus.

Pupillometry, as a measure of mental engagement, has the potential to be a valuable tool for the assessment of effort involved in speech processing [9,10]. Pupils respond to different stimuli; they constrict in response to brightness and fixation and dilate in response to increases in mental effort. Although pupil responses are not fully understood, pupil responses are similar to other eye movements, such as saccades, and responses have properties of reflexive and voluntary action. Such a tool is especially important for hearing-impaired (HI) individuals because effort can limit hearing rehabilitation, and effort management could become part of the diagnosis protocol. The heterogeneity within the clinical population of CI individuals, however, is often increased due to factors that relate to individual etiology and its resulting physiological changes in the auditory and speech neural systems, as well as to features that relate to hearing devices [11,12]. To ensure the external and internal validity of effort measurements within CI populations, we need to account for higher inter- and intraindividual variability in response to task demands, in particular, for tasks that depend on multiple processing stages, as does speech comprehension. Pupillometry has been used as an objective measure of mental effort for decades [13]. The strength of pupillometry is its physiological character, which makes the method objective as pupil dilation is beyond participants’ conscious control. Pupillometry data are often aggregated into measures of central tendency to characterize performance differences between groups, such as native versus nonnative listeners [14] or young people versus old people [15].

Cochlear implant electrodes are the boundary between the electrical stimulus and the auditory nerve that must be stimulated. Electrical impedance must be minimized in order to ensure an optimal stimulation. This parameter depends on different factors as: electrode surface or morphological and electrochemical processes initiated by the electrical stimulation. Moreover, electrode impedance provides an indication of the status of the electrode–tissue interface. Implant systems contain a measurement amplifier that can be configured to measure the intra-cochlear voltage field that results from the stimulation of a single-electrode pair along the entire electrode array. The full set of electrical spread curves, when normalized by the current, is known as trans-impedance matrix (TIM). Several studies have demonstrated the utility of TIM and other voltage measurement algorithms [16,17] to identify the intra-cochlear electrode positioning. The proximity of the electrode to the modiolar wall may result in some variation in the tissue and fluid environment, and it is of interest to investigate the effect of the modiolar proximity, the measured transimpedance and the cognitive effort on pitch discrimination.

It seems reasonable to suggest that perimodiolar position may be beneficial to improve speech discrimination in CI patients. The present study aims to establish the relation between these variables and whether perimodiolar positions provide statistically significant benefits regarding improved pitch discrimination, since there is a correlation between performance and electrode discrimination [18,19,20,21].

## 2. Materials and Methods

Participants: 24 post-lingual deaf adults were unilaterally implanted, with at least one year of use of the implant and good audiological performance (score > 70% Speech discrimination). The patients met the following inclusion criteria: they were adults (older than 18 years); diagnosed with severe-profound post-lingual progressive bilateral hearing loss (average pure tone audiometry thresholds higher than 70 dB) and post-lingual progressive hearing loss; they did not suffer from retrocochlear conditions, nor central auditory processing disorders; and patent and normal cochlear and nerve anatomy were present in all cases.

All patients were hearing aid users preoperatively and, in all cases, a disyllabic test < 40% was used as CI indication [22]. Twelve of them received a slim perimodiolar electrode array (CI632©Cochlear Nucleus) and 12 of them received a straight electrode array (CI622©Cochlear Nucleus). Nine of them were women and 15 were men, and their age ranged from 26 to 75 years old (mean 41.38 years).

Surgery proceeded without complications and complete electrode insertion was achieved in all patients; a Round Window (inferiorly enlarged) approach was used without Scala Vestibuli translocations. The postoperative location was checked by using Cone-Beam CT (MiniCat IQ, Xoran Technologies LLC, Ann Arbor, MI, USA); All patients had a stable map with 900 pps stimulation rate and 25 µs pulse and a minimum of 18 operational channels. A minimum of 12 months follow up was analyzed. All cases presented a 70% or more of speech understanding for sentence tests in silence without lip-reading using the CI at 65 dB HL.

Informed consent was obtained from all subjects involved in the study. The study was approved by the Ethics Committee of Hospital Universitario de Gran Canaria Doctor Negrín (protocol code 2020-405-1). 

### 2.1. Pitch Discrimination Method

The psychoacoustic study was performed by using a Psychoacoustic Research Platform designed in our Department. It was designed by using the Nucleus Implant Communicator library for Python (Python Software Foundation, v2.3). The researcher and patient interfaces were designed by using Visual Studio (Microsoft Corp. Visual Studio Community 2013, Redmond, WA, USA) and the stimulation scripts were built in Python to control the implant receiver/transmitter by sending instructions to a supplied cochlear implant research sound processor. The “Freedom” © processor was used in all cases.

This method has been previously described [23]. An alternative forced choice experiment using variable electrodes (e11, e12, e13, e14, e15), where the patient must guess which one of the three sounds is different, using two from variable electrode and one fixed electrode. Procedure: First define C&T levels for each electrode, then balance loudness at 25% of dynamic range and random electrode discrimination evaluation (three times) of each combination.

To minimize the effect of adaptation and learning, the stimulus parameters were set so that they were used by patients in their standard daily maps. Accordingly, the stimulation rate used for this test was 900 pps; the pulse width was 25 µs, the phase gap was 8 us and the stimulation mode was MP1 + 2. If such parameters had been changed, the speed at which each individual adapts to a new way of stimulation could have played a major role in electrode discrimination, thus hindering the effects sought out in this experiment.

### 2.2. Pupillometry Method

The pupillometry was performed by using a modified video-nistagmography system (Visual Eyes tm 515/525. Micromedical. Interacoustic, Chatham, IL, USA) with a light control system built-in in the goggles. For this purpose, custom-made software was developed by using MATLAB (R2008b, Math- Works, Natick, MA, USA) to detect the pupil and to calculate the dilatation diameter. This software was validated previously.

At the beginning of the test, a calibration of the pupil dilatation was performed in all patients to obtain the pupil dilatation dynamic range according to the power of the light source. Once the dilatation dynamic range was calculated, the light was set at 25% of the range (dim light condition). During the pitch discrimination test, the pupillometry system was synchronized to record the pupil dilatation 2 s before the stimuli presentation and 5 s after the stimuli presentation. Then, the maximum percent of dilatation, according to the pupil dilatation dynamic range, was calculated.

### 2.3. Radiological Study

Images were acquired by Cone-Beam computed tomography (CBCT) (MiniCat IQ, Xoran Technologies LLC, Ann Arbor, MI, USA). CBCT has been previously validated as a valuable tool for the assessment of electrodes post-cochlear implantation [24] as it requires less irradiation than a regular CT and it shows less sensitivity to metal artefacts. Thus, it represents an easier way to identify the electrode placement inside the cochlea.

From this image analysis, valuable information of the electrode position inside the cochlea can be obtained. We considered three measurements: Wrapping Factor, the Intracochlear Position Index, and the Homogeneity Factor, as previously described [25].

### 2.4. Wrapping Factor (WF)

The WF is one of the most valuable parameters to evaluate how perimodiolar an electrode array is. It indicates how tightly or loosely wrapped an electrode array is with respect to the modiolar wall. We use the wrapping factor measured according to Holden et al. [26]. Basically, the WF is the ratio between the inserted electrode length and the lateral wall length. WF is 1 when the electrode array is on the lateral wall, so the inserted electrode length and the lateral wall length is the same; therefore, the ratio is 1.0. On the other hand, WF value decreases as the array is wrapped more tightly to the modiolar wall.

### 2.5. Intracochlear Position Index (ICPI)

ICPI indicates the distance of each contact with respect to the modiolar wall. This measurement is normalized by electrode, being zero “0” the closest position to the modiolus and one “1” the closest position to the lateral wall.

### 2.6. Homogeneity Factor (HF)

HF provides information of the distance between each electrode and the modiolus. This measurement indicates the electrode array position with respect to the modiolus and the distance between each electrode and the modiolus. HF value of 0 means that all the electrodes of the array are placed at the same distance from the modiolus.

### 2.7. TransImpedance Matrix (TIM)

TIM is an objective parameter that can be configured to measure the intra-cochlear voltage field that results from the stimulation of one electrode pair along the entire electrode array, and the full set of electrical spread curves are normalized by the current. The measures are affected by many different variables, such as the electrode position with respect to the modiolar wall. On this study, the TIM is normalized in a range 0–1 for each patient and then, the volume under the surface is calculated.

TIM procedure is as follows: an electrode is stimulated and the observed electric potential along the electrode array is recorded. The neighbor electrode is then selected for stimulation and the next set of observations are recorded. This process is automatically repeated until the whole electrode array has been stimulated. As the distance from the stimulating electrode increases, the potential values decrease.

The applied signal to the electrodes is a biphasic square signal, the amplitude level of which is settled at 200 current levels, corresponding to 648 μA. The voltage values are fixed at 0–10.4 V. The determinations are performed at the end of the first trailing edge of the biphasic current pulse. The data are recorded in a “×” matrix. The rows define the target electrode, where the measurement is taken, and the columns refer to the active electrode, where the stimulus is produced.

A maximum value will be obtained on the stimulating electrode and a minimum value on the furthest electrode, although a great variability can be observed. The electric potential decays exponentially with distance, so the relative position of the electrodes can be correlated with the measured potential, in order to deduce the relative position between them. It is important to consider that other parameters like conductivity and cochlear geometry can contribute to this decay. If the other factors are not considered, the decay only indicates the proximity of the other electrodes, but not their exact distance between them.

### 2.8. Statistical Analysis

The statistical analysis of this study was performed by using IBM SPSS Statistics for Windows software v 24.0. A Kolmogorov–Smirnov test was performed in all cases to confirm that the data collected with WF, HF, ICPI, and electrode discrimination and pupillometry results come from a normal distribution. A one-way ANOVA test was used to compare the means of the CI groups. The t-test was used to compare between pairs to identify the different cases. The hypothesis contrast will be considered statistically significant when the corresponding “*p*” value is lower than 0.05.

## 3. Results

There were no statistical differences in disyllabic tests in both groups after 1 year follow-up (Table 1). No demographic differences between the CI622 and CI632 groups were found.

### 3.1. Electrode Discrimination Test

The electrode discrimination test was performed on each subject. The average success rate for the CI622 group was 41.6% (11–69%), and for the CI632 group was 75% (59–93%) (The percentage indicates the number of correct responses obtained during the psychoacoustic test) (*p* = 0.001) (Figure 1).

### 3.2. Radiological Studies

In all subjects, a postoperative image analysis was performed to obtain information about WF, ICPI, and HF. The average WF values were obtained from Table 2: for subjects implanted with CI622 was 0.86 and for subjects implanted with CI632 was 0.57 (*p* < 0.001). The ICPI value for subjects implanted with CI622 was 0.62 and for subjects implanted with CI632 was 0.08 (*p* < 0.001). Finally, the average HF value for subjects implanted with CI622 was 0.29 and for subjects implanted with CI632 was 0.06, (*p* < 0.001).

### 3.3. Pupillometry Test

The pupillometry test shows a higher pupil dilatation during the test on CI622 patients compared to CI632 (Figure 2). The pupil diameter (PD) increment from rest state (before the stimulus) on CI622 increase in 25.5% of the PD, and in patients with a CI632 the increment observed was 13.6% of the PD (*p* < 0.015) (Figure 3).

### 3.4. T.I.M.

All cases present normal position of the perimodiolar electrode, the intracochlear voltage drops monotonically as the distance between stimulation and recording contact increases, both toward the apex and toward the base. The relation between distance and absolute voltage drop is nonlinear, depending on the dimensionality and the material properties of the homogeneous medium. Details of the intracochlear potential map, such as peak levels and slopes, show substantial inter-subject variability in both groups, being more stable in CI632 group (not statistical difference, Figure 4). In this study, taken into account the intracochlear position of the CI632 electrode array, the apical contacts are placed relatively close to each other, also must be considered that in that area we found a cochlear duct very narrow, so it can be observed that the closer to modiolus it must be observed an increase of the impedance (Figure 5). Similar effect is observed in straight electrode (Figure 6). This finding is also observed in other theorical studies [27].

The improvement on electrode discrimination results observed in the CI632© group was 30% higher than the results observed in CI622©. Also, the pupil dilatation is 11.9% higher in the CI622© compared to CI632©, which means, a higher central process effort in straight electrode users (Table 2).

## 4. Discussion

It is assumed that subject’s ability to discriminate stimulation of one contact from another, will be related to the subject’s ability to understand speech and other environmental signals using the CI [28,29]. The cognitive effort for listening decreases as the discrimination between electrodes is easier. That has a big impact on the patient hearing workload, and higher workload means fatigue at the end of the day. Several studies have demonstrated that removing poorly discriminated electrodes from a subject’s processor map resulted in improved speech recognition scores in some subjects [23].

The two main objectives of this study were: first, to determine if TIM recording can provide information about perimodiolar or lateral position of the electrode and the relation with electrode discrimination; and, second, to evaluate the listening cognitive effort of perimodiolar or lateral wall electrode in electrode discrimination test.

As previous publications show [23,30], the electrode distance influences electrode discrimination. In this research, we also provide more empirical evidence in support of the hypothesis that the electrode distance to the inner wall of the cochlea is a significant variable for the prediction of electrode discrimination ability. Based on data from this study about WF, HF, and ICPI values, better results were obtained with the Nucleus CI632 than with the Nucleus CI622 according to their modiolar proximity and better intra-cochlear position.

It is not surprising to continually be reminded by patients and clinicians alike that conventional standardized speech perception measures do not fully describe real-life difficulties of CI patients. Speech communication includes multiple talkers, spatial locations, conversation topics, and distracting noises and, also, to be engaged in other activities like thinking about upcoming conversation topics, remembering recent events, and planning their responses. Pupillometry is a novel tool to evaluate the listening effort, it is an additional test to evaluate how easy or difficult is for these patients to participate in a conversation or under difficult hearing conditions.

It has been observed that patients implanted with CI532 and CI632 present a faster acquisition of auditory results compared with other implants. This is also an additional effect of less listening effort stimulation. The easier the hearing, the faster the acquisition of the benefit [31].

Regarding TIM results, the electric potential decays with distance in CI632©, so the relative position of the electrodes can be correlated with the measured potential, in order to deduce the relative position between them. It is important to consider that other parameters like conductivity and geometry can contribute to this decay. If the other factors are not considered, the decay only indicates the proximity of the other electrodes but not their exact distance between them. In general, the decay constants in TIM depend on electrode type and location, cochlear anatomy, and tissue properties. In general, we have observed a more stable situation of the impedance distribution along the electrode array when the electrodes are better positioned. In the cases with an increase distance between contacts and modiolar wall, an increased impedance has been observed, mainly in the apical region of the cochlea when compared with the basal turn [27].

Electrode distance to the modiolus correlates inversely with the success rate. Thus, distance to the inner wall will result in more difficulties to perceive electrode differences. It is important to notice that the more apical electrodes are usually placed closest to the modiolar wall due to a reduction of the diameter of Scala Tympani [21,32].

For straight electrode arrays, is important to consider reducing the number of contacts in order to avoid channel interactions [28,29]. On the other hand, as proposed in previous studies, the best way to solve channel interactions is to turn off a “poor” electrode [33].

As a weak aspect to consider in this study we have to mention that we used different types of electrode design, so we have to consider that the different electrode surface area, half-band vs. full-band electrode, might have influenced the results.

## 5. Conclusions

Significant differences in all different imaging measurement, pupillometry, and electrode discrimination results were observed. The TIM did not show statistical differences between the groups. The perimodiolar CI632 has a better electrode discrimination and a lower cognitive listening effort compared with the lateral wall electrode CI622. The closer the electrode position to the modiolar wall, the better the electrode discrimination and the lower the listening effort.

## Figures and Tables

**Figure 1 jcm-10-03962-f001:**
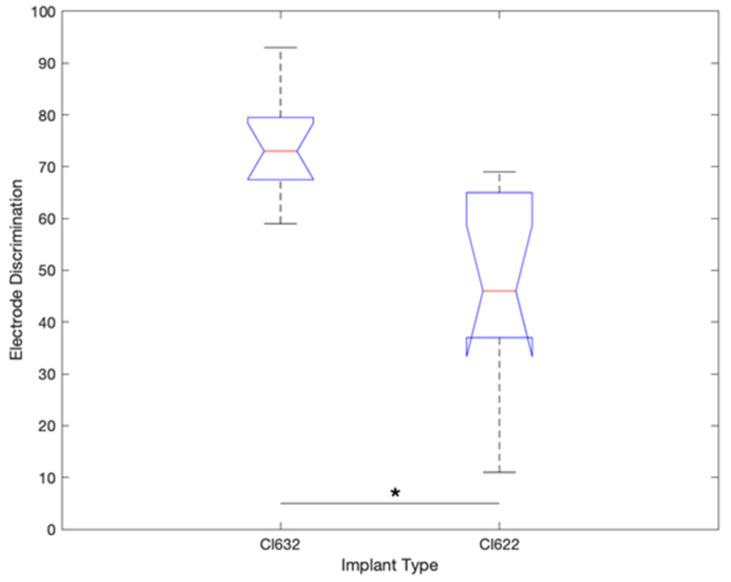
Electrode discrimination test result comparison chart between CI632 and CI622 recipients (* *p* < 0.05).

**Figure 2 jcm-10-03962-f002:**
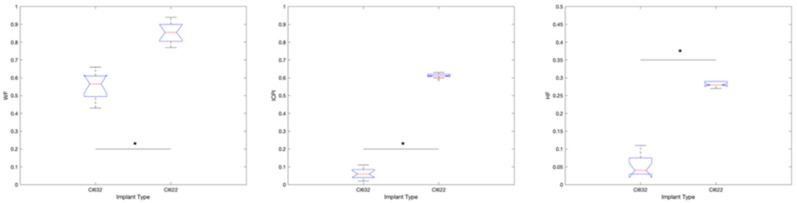
Radiological evaluation comparison chart between CI632 and CI622 recipients (* *p* < 0.05).

**Figure 3 jcm-10-03962-f003:**
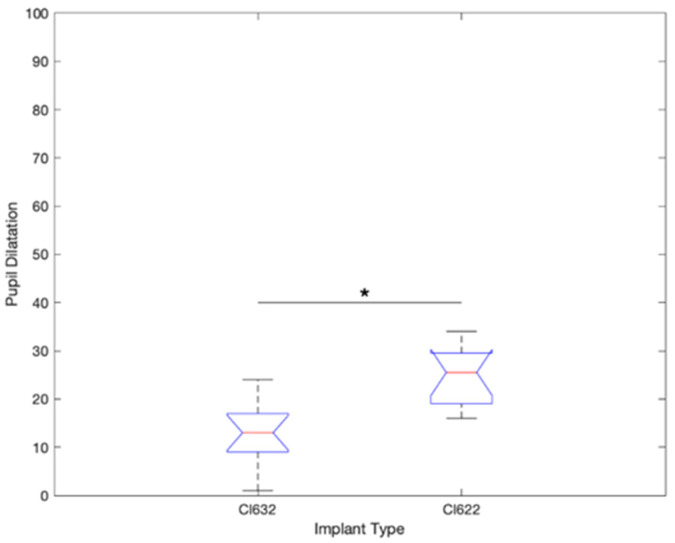
Pupil dilatation result comparison chart between CI632 and CI622 recipients during electrode discrimination test (* *p* < 0.05).

**Figure 4 jcm-10-03962-f004:**
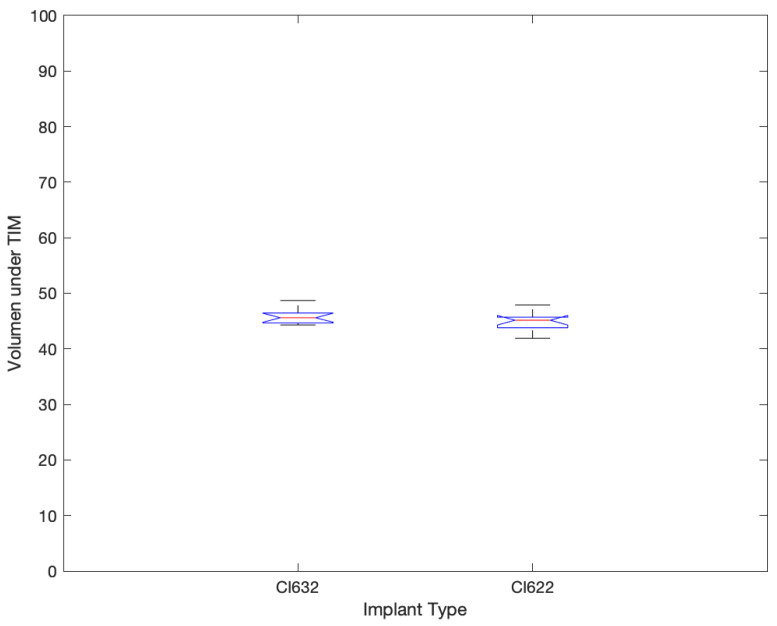
Volume under TIM result comparison chart between CI632 and CI622 recipients.

**Figure 5 jcm-10-03962-f005:**
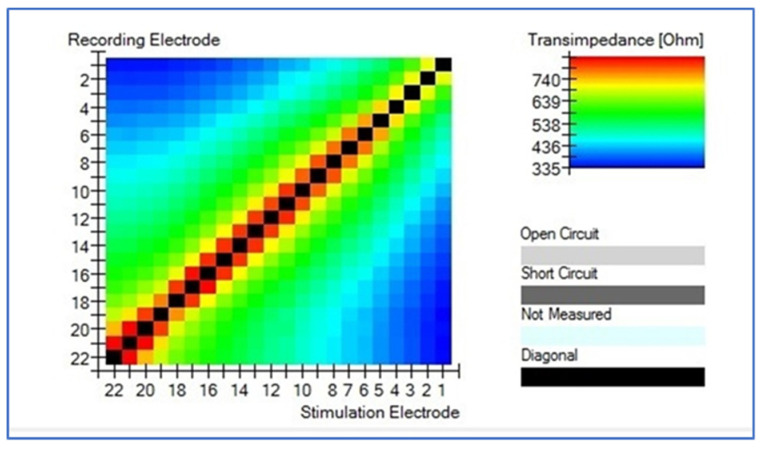
Typical transimpedance measurement result. The intracochlear voltage drops monotonically as the distance between stimulation and recording contact increases, both toward the apex and toward the base, in perimodiolar electrode cases (CI632). HEAT map and Line graph are presented.

**Figure 6 jcm-10-03962-f006:**
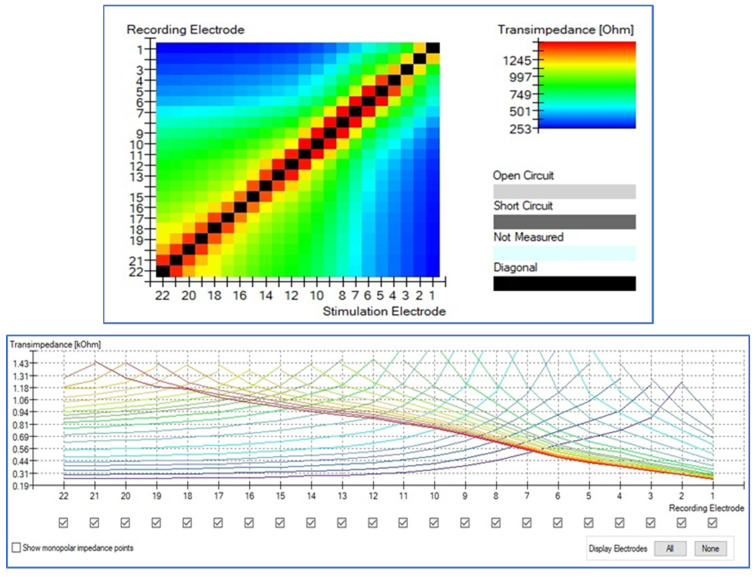
Typical transimpedance measurement result. The intracochlear voltage drops monotonically as the distance between stimulation and recording contact increases, both toward the apex and toward the base, in perimodiolar electrode cases (CI622). HEAT map and Line graph are presented. Higher impedances are observed.

**Table 1 jcm-10-03962-t001:** Patients’ demographic data.

Subject	Type of Implant	Sex	Age (Years)	Prosthesis Use (Month)	Tinnitus	Number of Active Electrodes	Preop. Deafness Duration (Years)	Disyllabics %	Implant Ear
S1	CI622	F	64	16	No	22	4	92	R
S2	CI622	F	26	29	No	22	7	76	R
S3	CI622	M	53	48	No	22	5	72	R
S4	CI622	M	70	20	Yes	22	6	80	L
S5	CI622	F	52	34	No	22	5	80	R
S6	CI622	M	34	27	No	22	8	84	L
S7	CI622	M	75	43	No	22	9	96	L
S8	CI622	F	66	16	No	22	12	96	L
S9	CI622	M	60	30	No	22	6	92	R
S10	CI622	M	32	36	Yes	22	1	76	R
S11	CI622	M	49	43	No	22	4	80	L
S12	CI622	F	57	48	No	22	11	92	R
S13	CI632	M	58	31	No	22	3	68	L
S14	CI632	F	56	12	No	22	9	96	L
S15	CI632	F	47	21	No	22	11	88	R
S16	CI632	M	30	17	No	22	1	92	L
S17	CI632	M	58	58	No	22	6	96	L
S18	CI632	F	28	22	No	22	3	88	L
S19	CI632	M	50	38	Yes	22	15	96	R
S20	CI632	F	75	27	Yes	22	9	88	R
S21	CI632	M	66	40	No	22	11	84	L
S22	CI632	M	70	20	No	22	3	88	R
S23	CI632	M	49	13	No	22	5	92	L
S24	CI632	M	47	12	No	22	2	88	L

**Table 2 jcm-10-03962-t002:** Correlation for radiological, electrode discrimination, pupil dilatation, and volume under the TIM (TransImpedance Matrix) surface results.

Subject	Implant	WF	ICPI	HF	Electrode Discrimination (%)	Pupil Dilatation (%)	Volume under TIM Surface
S1	CI622	0.94	0.60	0.29	46	28	46.7
S2	CI622	0.77	0.62	0.28	64	28	45.4
S3	CI622	0.87	0.61	0.29	66	19	46.0
S4	CI622	0.81	0.62	0.28	69	25	45.4
S5	CI622	0.89	0.62	0.29	11	24	42.4
S6	CI622	0.80	0.61	0.28	29	31	45.4
S7	CI622	0.91	0.61	0.28	46	34	44.3
S8	CI622	0.87	0.63	0.28	46	34	44.0
S9	CI622	0.84	0.59	0.29	45	19	43.6
S10	CI622	0.79	0.61	0.27	66	26	44.9
S11	CI622	0.91	0.61	0.28	30	16	41.9
S12	CI622	0.82	0.62	0.28	44	17	47.9
S13	CI632	0.47	0.11	0.11	78	14	47.0
S14	CI632	0.43	0.02	0.03	67	15	45.9
S15	CI632	0.66	0.03	0.04	59	10	45.9
S16	CI632	0.57	0.06	0.03	93	16	44.5
S17	CI632	0.56	0.05	0.06	78	18	48.0
S18	CI632	0.66	0.06	0.03	71	24	45.7
S19	CI632	0.62	0.11	0.10	85	12	44.5
S20	CI632	0.60	0.10	0.09	72	1	48.7
S21	CI632	0.58	0.07	0.04	81	8	45.5
S22	CI632	0.56	0.07	0.05	74	22	44.8
S23	CI632	0.51	0.03	0.03	68	7	45.4
S24	CI632	0.48	0.05	0.04	64	10	44.3
CI622	Mean	0.8517	0.6125	0.2825	46.83	25.08	44.82
	Std. Dev.	0.0546	0.0106	0.0062	17.6318	6.2879	1.7121
CI632	Mean	0.5583	0.0633	0.0542	74.16	13.0833	45.85
	Std. Dev.	0.0736	0.0306	0.0294	9.466	6.5012	1.4003
Anova Test	*p*	1.77^−10^	1.06^−25^	3.93^−18^	0.0001	0.0001	0.1227
	F	123.01	2464.17	693.98	22.39	21.12	2.58
	Degrees of freedom	1	1	1	1	1	1

## Data Availability

The data presented in this study are available on request from the corresponding author. The data are not publicly available due to informed consent restrictions.

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
