# Peer review of "Analysis of Neural Interface When Using Modiolar Electrode Stimulation. Radiological Evaluation, Trans-Impedance Matrix Analysis and Effect on Listening Effort in Cochlear Implantation"

_jcm, 2021, doi:10.3390/jcm10173962_

Round 1

Reviewer 1 Report

Overall, the study presents novel and interesting findings regarding the relationship between (a) the type of electrode and position of the electrode array and (b) listening effort and hearing resolution, using behavioral and objective measures. This data may be of great interest to the readers of JCM. In its current version, however, the introduction, rational for the study and for choosing the specific tests, and the discussion are hard to follow. An extensive reorganization and editing of the Introduction and Discussion is needed for the manuscript to be suitable for publication. In addition, I detailed a list of comments that I would appreciate if the authors could address.

Title

The title is somewhat misleading as the electrode position was evaluated using mainly the radiology results. I would suggest rephrasing the title, and maybe including also the effect on listening effort.

Abstract

  1. "A descriptive prospective randomized study" – I believe that this study is not a descriptive study.
  2. "better stimulation (electrode discrimination) and better mental effort (pupillometry)". – please rephrase. Consider "better hearing resolution and less listening effort"
  3. "patients implanted with peri-modiolar electrode obtain… and more homogenous TIM patterns" – This line is somewhat misleading as in the results section you reported that no significant differences were shown in TIM between the two groups.
  4. "Trans-Impedance Matrix seems to be a useful tool to improve the better positioning of the perimodiolar array. A narrower excitation pattern, may provide a better stimulation in cochlear implant recipients." – This conclusion cannot be drawn from the current results as you didn’t show a significant correlation between the TIM and radiology measures or pitch discrimination.

Introduction

  1. It is hard to follow the introduction as it is not clear how one paragraph leads to the other.
  2. "Pupillometry, as a measure of mental engagement, has the potential to be a valuable tool for the assessment of effort involved in speech processing" – References needed.
  3. A general explanation regarding the Pupillometry test is missing.
  4. "The present study aims to … and establish whether perimodiolar positions provide statistically significant benefits regarding improved pitch discrimination" - This is the first time that the term "perimodiolar" is mentioned in the introduction. Please introduce the term earlier and explain why do you hypothesize that "it seems reasonable to suggest that, perimodiolar position may be beneficial to improve speech discrimination in CI patients ".

Materials and methods

  1. The first paragraph in the introduction should be in the Materials and Method section, under Participants.
  2. Table 1- "deafness duration" – is it before implantation? Please specify
  3. Table 1- There is some information missing: 1) as the participants were sequentially implanted, how much time was it between the first and second implant? For how long did they use hearing aids in each ear before implantation? What was the Disyllabic score before implantation?
  4. Were the subjects tested with both implants? table 1 contains data regarding a single implant for each subject…
  5. Were there any demographic differences (implantation age, duration of deafness, speech scores before implantation etc.) between the CI622 and CI632 groups? Even if there were no significant differences it is worth mentioning.
  6. "and, finally, an alternative forced choice experiment was completed". – Please provide an explanation regarding the paradigm of the pitch discrimination test. What were the instructions to the participants, was this an adaptive procedure? How were the thresholds calculated and so on…?
  7. Statistical analysis – Please state in this section were there any corrections for multiple comparisons?

Results

  1. Please provide the full statistical result for each analysis (in addition to p values please write F values, degrees of freedom, effect sizes etc.)
  2. Please add a figure that compares the radiological results between the two groups (similarly to the pitch discrimination and pupillometry results)
  3. "A statistical analysis between both groups on each different imaging measurement, pupillometry, TIM and the electrode discrimination results was performed. Statistical Bonferoni corrected t test shows significant differences in all measurements except TIM: WF (p < 0.001), HF (p < 0.001), ICPI (p < 0.001), pitch discrimination (p = 0.015) and pupil-lometry (p = 0.015). For the volume under the surface generated by TIM there is no statis-tical differences between both populations (p = 0.415)." – This paragraph seems redundant as most of this information is described earlier.
  4. Please test whether there are significant correlations between the radiological, electrode discrimination, pupil dilatation and volume under the TIM surface results shown in Table 2.

Author Response

Review 1

We would like to thank the revision made by the expert. We have modified the manuscript according to his/her recommendations.

Overall, the study presents novel and interesting findings regarding the relationship between (a) the type of electrode and position of the electrode array and (b) listening effort and hearing resolution, using behavioral and objective measures. This data may be of great interest to the readers of JCM. In its current version, however, the introduction, rational for the study and for choosing the specific tests, and the discussion are hard to follow. An extensive reorganization and editing of the Introduction and Discussion is needed for the manuscript to be suitable for publication. In addition, I detailed a list of comments that I would appreciate if the authors could address.

Global Changes have been made

Title

The title is somewhat misleading as the electrode position was evaluated using mainly the radiology results. I would suggest rephrasing the title, and maybe including also the effect on listening effort.

The title has been changed.

Analysis of neural interface when using modiolar electrode stimulation. Radiological evaluation, Trans-Impedance Matrix analysis, and effect on listening effort  in cochlear implantation.

Abstract

  1. "A descriptive prospective randomized study" – I believe that this study is not a descriptive study. Changed .
  2. "better stimulation (electrode discrimination) and better mental effort (pupillometry)". – please rephrase. Consider "better hearing resolution and less listening effort" Changed .
  3. "patients implanted with peri-modiolar electrode obtain… and more homogenous TIM patterns" – This line is somewhat misleading as in the results section you reported that no significant differences were shown in TIM between the two groups. Changed to : improve the better  analyze positioning
  4. "Trans-Impedance Matrix seems to be a useful tool to improve the better positioning of the perimodiolar array. A narrower excitation pattern, may provide a better stimulation in cochlear implant recipients." – This conclusion cannot be drawn from the current results as you didn’t show a significant correlation between the TIM and radiology measures or pitch discrimination. Removed the sentence

Introduction

  1. It is hard to follow the introduction as it is not clear how one paragraph leads to the other.

We have modified the text

  1. "Pupillometry, as a measure of mental engagement, has the potential to be a valuable tool for the assessment of effort involved in speech processing" – References needed.

Hornsby, B. W. (2013). The effects of hearing aid use on listening effort and mental fatigue associated with sustained speech processing demands. Ear and Hearing, 34(5), 523–534.

Chapman, L. R., & Hallowell, B. (2015). A novel pupillometric method for indexing word difficulty in individuals with and without aphasia. Journal of Speech, Language, and Hearing Research, 58(5), 1508–1520.

  1. A general explanation regarding the Pupillometry test is missing.

We have included: “Pupils respond to different stimuli, they constrict in response to brightness and fixation and dilate in response to increases in mental effort. Although pupil responses are not fully understood, pupil responses are similar to other eye movements, such as saccades and responses have properties of reflexive and voluntary action.”

  1. "The present study aims to … and establish whether perimodiolar positions provide statistically significant benefits regarding improved pitch discrimination" - This is the first time that the term "perimodiolar" is mentioned in the introduction. Please introduce the term earlier and explain why do you hypothesize that "it seems reasonable to suggest that, perimodiolar position may be beneficial to improve speech discrimination in CI patients ".

We have added the sentence proposed by the reviewer.

Materials and methods

  1. The first paragraph in the introduction should be in the Materials and Method section, under Participants. Changed

  1. Table 1- "deafness duration" – is it before implantation? Please specify Changed

  1. Table 1- There is some information missing: 1) as the participants were sequentially implanted, how much time was it between the first and second implant? For how long did they use hearing aids in each ear before implantation? What was the Disyllabic score before implantation?

All data were mentioned

It is impoprtant that the translation of “sequentilally” is wrong as it refers to “consecutive implanted”. All cases were unilateral CI ( modified)

  1. Were the subjects tested with both implants? table 1 contains data regarding a single implant for each subject…

Explained in the previous comment

  1. Were there any demographic differences (implantation age, duration of deafness, speech scores before implantation etc.) between the CI622 and CI632 groups? Even if there were no significant differences it is worth mentioning.

Modified

  1. "and, finally, an alternative forced choice experiment was completed". – Please provide an explanation regarding the paradigm of the pitch discrimination test. What were the instructions to the participants, was this an adaptive procedure? How were the thresholds calculated and so on…?

We include the text: “This method has been previously described [21]. An alternative forced choice experiment using variable electrodes (11, 12, 13, 14, 15), and patient must guess which one of the 3 sound is different, using two from variable electrode and one fixed electrode. Procedure: First define C&T levels for each electrodes, then balance loudness at 25% of dynamic range and random electrode discrimination evaluation (3 times) of each combination.”

Results

  1. Please provide the full statistical result for each analysis (in addition to p values please write F values, degrees of freedom, effect sizes etc.)

INCLUDED

  1. Please add a figure that compares the radiological results between the two groups (similarly to the pitch discrimination and pupillometry results)

INCLUDED

  1. "A statistical analysis between both groups on each different imaging measurement, pupillometry, TIM and the electrode discrimination results was performed. Statistical Bonferoni corrected t test shows significant differences in all measurements except TIM: WF (p < 0.001), HF (p < 0.001), ICPI (p < 0.001), pitch discrimination (p = 0.015) and pupil-lometry (p = 0.015). For the volume under the surface generated by TIM there is no statis-tical differences between both populations (p = 0.415)." – This paragraph seems redundant as most of this information is described earlier.

Paragraph has been excluded according to  reviewer recommendation

  1. Please test whether there are significant correlations between the radiological, electrode discrimination, pupil dilatation and volume under the TIM surface results shown in Table 2.

Final Comment:

We have reduced the similarity below to 5% or lower by rephrasing or citing the source instead.

English has been reviewed by a native expert.

Reviewer 2 Report

Overall I read a great manuscript. Just a few comments:

Introduction:

It is said that „several studies have demonstrated the utility of TIM…“ but just one is cited. Please add further.

Material and Methods:

The pitch discrimination method might be explained in more detail. There is a citation to find out more but it would help the reader to get some more information in the text.

Pupillometry method … „de“… typing error.

The Wrapping factor should be explained in more detail. How do you get a number out of it?

Statistical Analysis … p in italics

Results:

  • Mean of years „yo“ change into „y“ or „years“
  • Please give some information to the dysillabic test for international readers
  • TIM is a sceenshot. How do you work with the individual TIMs. How is the comparison between patients done? Could you provide at least to TIMs (one with 632 and one with 622)?

Discussion:

The point half-band vs. Full-band is addressed very late. Please put this in the introdurction, too.

Author Response

Review 2

We thank for the comments

Overall I read a great manuscript. Just a few comments:

Introduction:

It is said that „several studies have demonstrated the utility of TIM…“ but just one is cited. Please add further.

We have added a recent paper: Klabbers TM, Huinck WJ, Heutink F, Verbist BM, Mylanus EAM. (2021)Transimpedance Matrix (TIM) Measurement for the Detection of Intraoperative Electrode Tip Foldover Using the Slim Modiolar Electrode: A Proof of Concept Study.Otol Neurotol. 1;42(2):e124-e129.

Material and Methods:

The pitch discrimination method might be explained in more detail. There is a citation to find out more but it would help the reader to get some more information in the text.

We have included: “This method has been previously described [21]. An alternative forced choice experiment using variable electrodes (11, 12, 13, 14, 15), and patient must guess which one of the 3 sound is different, using two from variable electrode and one fixed electrode. Procedure: First define C&T levels for each electrodes, then balance loudness at 25% of dynamic range and random electrode discrimination evaluation (3 times) of each combination.”

Pupillometry method … „de“… typing error.

Correction made.

The Wrapping factor should be explained in more detail. How do you get a number out of it?

We include for more information the Holden paper which describes the technique. Holden LK, Finley CC, Firszt JB, Holden TA, Brenner C, Potts LG, Gotter BD, Vanderhoof SS, Mispagel K, Heydebrand G, Skinner MW.(2013)Factors affecting open- set word recognition in adults with cochlear implants. Hear Hear. 34:342–360.

Statistical Analysis … p in italics

Correction made

Results:

  • Mean of years „yo“ change into „y“ or „years“
  • Please give some information to the dysillabic test for international readers
  • TIM is a sceenshot. How do you work with the individual TIMs. How is the comparison between patients done? Could you provide at least to TIMs (one with 632 and one with 622)?

We have change the errors , include the 622 f¡igure and include the citation :  Huarte A (2008) The Castilian Spanish Hearing in Noise Test . Int JAudiol 47(6):369-7022 , for dysillabic word test used.

Final Comments:

We have reduced the similarity below to 5% or lower by rephrasing or citing the source instead.

English has been reviewed by a native expert.

Round 2

Reviewer 1 Report

The authors wrote in their cover letter that "It is important that the translation of “sequentilally” is wrong as it refers to “consecutive implanted”. All cases were unilateral CI ( modified)

However, in the Participants' section it is still written "24 post-lingual deaf adults were sequentially implanted".  Instead of unilaterally implanted. Please change.

Furthermore, all the data regarding the background information of the participants that is included in the Results section: "24 post-lingual deaf adults were unilaterally implanted, all patients were hearing aid users preoperatively and, in all cases, a disyllabic test < 40% was used as CI indication. (26) 12 of them received a slim perimodiolar electrode array (CI632©Cochlear Nucleus) and 12 of them received a straight electrode array (CI622©Cochlear Nucleus). 9 of them were women and 15 were men, and their age ranged from 26 to 75 years old (mean 41,38 y). There were no statistical differences in disyllabic tests in both groups after 1 year follow-up (Table 1). No demographic differences between the CI622 and CI632 groups were found." Should be transferred to the Participants section.

English editing is required for the modified text in the new version of the manuscript.

Author Response

Reviewer 2

First of all we would like to thank the reviewer for the interest and comments.

1.The authors wrote in their cover letter that "It is important that the translation of “sequentilally” is wrong as it refers to “consecutive implanted”. All cases were unilateral CI ( modified)However, in the Participants' section it is still written "24 post-lingual deaf adults were sequentially implanted".  Instead of unilaterally implanted. Please change.

We have change the text, according to reviewer´s recommendation

Furthermore, all the data regarding the background information of the participants that is included in the Results section: "24 post-lingual deaf adults were unilaterally implanted, all patients were hearing aid users preoperatively and, in all cases, a disyllabic test < 40% was used as CI indication. (26) 12 of them received a slim perimodiolar electrode array (CI632©Cochlear Nucleus) and 12 of them received a straight electrode array (CI622©Cochlear Nucleus). 9 of them were women and 15 were men, and their age ranged from 26 to 75 years old (mean 41,38 y). There were no statistical differences in disyllabic tests in both groups after 1 year follow-up (Table 1). No demographic differences between the CI622 and CI632 groups were found." Should be transferred to the Participants section.

The text has been modified and transferred to Participants Section

English editing is required for the modified text in the new version of the manuscript.

English of the modified text has been reviewed.